# *In* *Vitro* Anthelmintic Activity of Sea Buckthorn (*Hippophae rhamnoides*) Berry Juice against Gastrointestinal Nematodes of Small Ruminants

**DOI:** 10.3390/biology11060825

**Published:** 2022-05-27

**Authors:** Michela Maestrini, Cristina Forzato, Simone Mancini, Ylenia Pieracci, Stefania Perrucci

**Affiliations:** 1Department of Veterinary Sciences, University of Pisa, Viale delle Piagge 2, 56124 Pisa, Italy; simone.mancini@unipi.it; 2Department of Chemical and Pharmaceutical Sciences, University of Trieste, Via L. Giorgieri 1, 34127 Trieste, Italy; cforzato@units.it; 3Department of Pharmacy, University of Pisa, Via Bonanno Pisano 6, 56126 Pisa, Italy; ylenia.pieracci@phd.unipi.it

**Keywords:** gastrointestinal nematodes, sheep, *Hippophae rhamnoides*, anthelmintic activity, polyphenolic compounds, isorhamnetin, quercetin

## Abstract

**Simple Summary:**

Gastrointestinal nematodes are included worldwide among the most prominent parasites of small ruminants. In past decades, the control of these nematodes mainly relied on the use of synthetic anthelmintic drugs. However, nowadays the exclusive use of anthelmintic drugs is considered an obsolete and unsustainable control strategy due to the onset of anthelmintic-resistant sheep gastrointestinal nematode strains and the issues linked with the environmental pollution and residues in food of animal origin of synthetic anthelmintic drugs. Among alternative or complementary methods, the use of plants endowed with anthelmintic properties has been identified as a valuable option. As a part of this approach, this study evaluated the *in vitro* anthelmintic properties of two commercial sea buckthorn (*Hippophae rhamnoides*) berry juices on sheep gastrointestinal nematodes. Both *H. rhamnoides* berry juices tested in this study showed interesting anthelmintic properties *in vitro*. The obtained results are promising regarding the use of sea buckthorn berry juice as a potential tool for the control of gastrointestinal nematodes in small ruminants.

**Abstract:**

Gastrointestinal nematodes are one of the major threats in small ruminant breeding. Their control is difficult due to the development of anthelmintic resistance, and the search for new molecules endowed with anthelmintic activity (AH) is considered a priority. In this context, we evaluated the *in vitro* AH activity of two commercial sea buckthorn (*Hippophae rhamnoides*) berry juices, namely SBT and SBF. The *in vitro* evaluation was based on the egg-hatch test and larval exsheathment assay at different concentrations. Data were statistically analysed, and the EC_50_ was calculated. Chemical analyses were performed to evaluate the total polyphenol content of the juices and chemical profile of the most represented compounds. The role of the polyphenolic fraction in the anthelmintic activity of the juices was also assessed. At the highest concentrations, the activity of SBT was high in both tests and comparable to that observed in the thiabendazole-treated positive controls, while SBF showed a lower efficacy. Glycosylated isorhamnetin and quercetin were the most represented polyphenolic compounds in both juices. In conclusion, both *H. rhamnoides* berry juices tested in this study showed interesting anthelmintic properties *in vitro*.

## 1. Introduction

Gastrointestinal nematodes (GIN) are one of the most concerning threats to welfare and productivity in small ruminant breeding. They may cause severe clinical signs and the death of infected animals, depending on the species involved, the intensity of the infection, and individual factors [1,2,3]. Every year, GIN cause a relevant damage in economic terms due to production losses and the cost of anthelmintic treatments [4,5]. In past decades, the control of GIN relied mainly on the regular use of synthetic anthelmintic drugs (ADs), often without previous parasitological analyses and using the same molecules over the years. This often-irrational drug management led to the worldwide widespread of GIN anthelmintic resistance [6,7,8]. ADs are also responsible for environmental contamination, since they are eliminated by the animal on the soil, and they can remain as residues in products of animal origin [9,10,11]. For these reasons, there is an important increased interest in organic products, and every day consumers become more aware of their eating habits, preferring safe and ethically produced foods [12,13]. In recent years, scientific research has investigated the potential anthelmintic properties of natural compounds, such as plant extracts and their metabolites, with the aim to find new anthelmintic molecules and ecofriendly solutions for the control of animal parasites [14,15,16,17].

In this context, this study was designed to evaluate the *in vitro* anthelmintic properties of two different berry juices of sea buckthorn (*Hippophae rhamnoides*), a well-known a medicinal plant [18,19]. In northern Europe, the therapeutic virtues of sea buckthorn have been known since the Middle Ages. In Scandinavian countries, for example, people knew of and used sea buckthorn berries not only in folk medicine, but also in the preparation of jams and dried fruit together with honey [20]. With varied bioactive and curative activities, it was traditionally used for the treatment of skin disorders, peptic ulcers, heart problems, tumors, cough, jaundice, asthma, hypertension, rheumatism, and inflammation of genital organs [18,19,21]. Antioxidative, immunomodulating [22], antibacterial, antiviral, antifungal [23], antidiabetic, and hepatoprotective activities [21,24] also have been reported. Nowadays, it can be found among the components of beauty products such as anti-stretch-mark creams and gels [25].

The *in vitro* anthelmintic properties of two commercial *H. rhamnoides* berry juices from two different Italian regions, Tuscany and Friuli-Venezia Giulia, were evaluated and compared in this study. Chemical analyses were performed to evaluate the total polyphenols content of the juices, and the identification of the most representative compounds was performed by using HPLC and HPLC-MS analyses. The role of polyphenols in the anthelmintic activity of the extracts was also estimated by adding polyvinylpolypyrrolidone (PVPP), a substance able to bind flavonoids and a variety of other phenolic compounds [26].

## 2. Materials and Methods

### 2.1. Chemicals

Folin–Ciocalteau reagent, gallic acid, quercetin, methanol, trifluoroacetic acid (TFA), Na_2_CO_3_, and acetonitrile (HPLC grade) were purchased from Sigma-Aldrich Chemie GmbH (Schnelldorf, Germany). Isorhamnetin, myricetin, and epicatechin were from TCI Europe N.V. (Zwijndrecht, Belgium). Isorhamnetin-3-glucoside, rutin, and gallocatechin were from Extrasynthese (Lyon, France). Epigallocatechin and kaempferol were from Alfa Aesar Thermo Fisher Scientific, (Haverhill, MA, USA). Polyvinylpolypyrrolidone (PVPP) and thiabendazole (2-(4-Thiazoly) benzimidazole) were purchased from Sigma-Aldrich S.r.l. (Milan, Italy).

### 2.2. Plant Material

Two different SB berry juices commercialized for human consumption were used. The first one (SBT) was from an organic farm (Succo e polpa di olivello spinoso, Demeter, Azienda Agricola San Mario) located in Bibbona (Livorno, Tuscany, Italy). This juice was frozen and lyophilized (Modulyo, Pirani 501, Edwards, UK) at the Department of Pharmacy, University of Pisa (Pisa, Italy) and opportunely stored. The other juice (SBF) was obtained from an organic farm (Succo madre olivello spinoso, Olispin S.S. Agricola), located in Mortegliano, (Udine, Friuli-Venezia Giulia, Italy). This juice was frozen and lyophilized (Edwards Modulyo freeze-dryer) at the Department of Chemical and Pharmaceutical Sciences, University of Trieste (Trieste, Italy) and opportunely stored.

### 2.3. Determination of Total Polyphenol Content

The total polyphenols content (TPC) of the juices was determined by using the Folin–Ciocalteau method [27], and the results were expressed as milligrams of gallic acid equivalents per liter of juice sample (mg GAE/L). A calibration curve with standard solutions of gallic acid was used in the range of 0.02–0.7 mg/mL. Solutions for the calibration curve were prepared by introducing 0.2 mL of standard solution of gallic acid, 1.2 mL of methanol, 0.5 mL of 2N Folin–Ciocalteu reagent, and 0.5 mL of 7.5% Na_2_CO_3_, respectively, into a 10 mL volumetric flask and adjusting the volume with distilled water. Similarly, a blank was prepared by replacing the gallic acid solution with 0.2 mL of methanol. The solutions were left to stand for 1 h in the dark, after which the absorbance at 765 nm was measured in a 1 cm cuvette against the reagent blank. The calibration curve for gallic acid (y = 0.8354x − 0.0012) showed good response linearity with a correlation coefficient (R^2^) of 0.9995. To determine the total polyphenols in sea buckthorn juices, a mother solution was prepared by placing 2.5 mL of each juice in a 25 mL volumetric flask and adjusting the volume with distilled water. The solutions for the spectrophotometric readings were prepared by introducing 0.2 mL of the mother solution of each sea buckthorn juice, 1.2 mL of methanol, 0.5 mL of 2N Folin–Ciocalteu reagent, and 0.5 mL of 7.5% Na_2_CO_3_ into a 10 mL volumetric flask and adjusting the volume with distilled water. Similarly, a blank was prepared by replacing the sea buckthorn solution with 0.2 mL of distilled water. The absorbance values were measured on a Perkin-Elmer UV/VIS Lambda 2 spectrometer.

### 2.4. HPLC and HPLC-MS Analyses

HPLC analyses were performed on a Vanquish Thermo Chromatograph with an autosampler and a diode-array detector at 280 and 360nm using a Kinetex C18 150 × 2.1 mm 5 µ 100 Å (Phenomenex, Torrance, CA, USA) column at 35 °C with a flow rate of 0.25 mL/min. The injection volume was 20 µL. The eluents used were A: H_2_O + 0.05% TFA and B: CH_3_CN + 0.05% TFA with the following gradient: t_0_ 97% eluent A + 3% eluent B, t_5min_ 97% eluent A + 3% eluent B, t_10min_ 95% eluent A + 5% eluent B, t_45min_ 65% eluent A + 35% eluent B, t_50min_ 0% eluent A + 100% eluent B, t_53min_ 97% eluent A + 3% eluent B, t_60min_ 97% eluent A + 3% eluent B.

The following standards were used for the identification of polyphenols in sea buckthorn juices: quercetin, isorhamnetin, isorhamnetin-3-glucoside, myricetin, rutin, epicatechin, gallocatechin, epigallocatechin, and kaempferol. Sample solutions were prepared using 200 µL of each juice and adding 1800 µL of Milli-Q water. The solution was filtered through a 0.46 µm PTFE syringe filter and injected into the HPLC.

The HPLC-MS analysis was performed on a HPLC Agilent 1260 Infinity II with an autosampler coupled with a mass detector using an ESI source micrOTOF-Q (Bruker, Billerica, MA, USA). Measurements were registered in positive mode. HPLC conditions were the same as described above.

Calibration curves with standard solutions of quercetin and isorhamnetin-3-glucoside for quantitative determination at 360 nm were obtained in the range of 4–120 µg/mL for quercetin and 10–540 µg/mL for isorhamnetin-3-glucoside. The calibration curves for quercetin (y = 1.2673x + 3.3152) and isorhamnetin-3-glucoside (1.7699x + 5.7356) showed good response linearity, with a correlation coefficient (R^2^) of 0.9997 and 0.9996, respectively. The LOD and LOQ values for quercetin were 0.06 µg/mL and 0.18 µg/mL, respectively; while for isorhamnetin-3-glucoside, they were 0.05 µg/mL and 0.14 µg/mL, respectively.

### 2.5. Recovery and Suspension of GIN Eggs and Third-Stage Larvae (L3), and Identification of GIN L3 at the Genus/Species Level

Individual faecal samples were collected from the rectum of naturally infected ewes. Ewes were infected by different GIN genera and species. Parasitological analysis of the collected samples was performed using the McMaster method with a sensitivity of 50 eggs per gram (EPG) of faeces [28]. Faecal samples scoring positive for at least 1000 EPG were pooled and used in the assays and for preparing the faecal cultures to obtain fresh L3. Recovery and suspension of the eggs were performed using a previously reported protocol [29] with small modifications. In short, 30 g of faecal material was homogenized in distilled water, placed inside a 50 mL tube, and centrifuged for 5 min at 2300 rpm. The sediment was collected and suspended in saturated NaCl solution (specific density 1.2) and centrifuged for another 5 min at 1000 rpm. The supernatant was then collected, diluted in distilled water in 15 mL tubes, and then centrifuged for 5 min at 800 rpm. The sediment containing the eggs was collected and diluted in 1 mL of distilled water for GIN eggs/mL determination. Faecal cultures were placed in an incubator at 27 °C from 7 up to 10 days. L3 were recovered by using the Baermann technique [30] and were used to perform the larval exsheathment inhibition assay (LEIA). Moreover, about 100 larvae were microscopically identified at the genus level based on their morphological and metric features [31]. In brief, L3 identification was based on several ensheathed L3 characteristics, including L3 dimensions (length and width), number and shape of intestinal cells, length and shape of the tail, shape of the head, the presence or absence of cranial refractile spots, and length and shape of the oesophagus. The presence or absence of digitate appendages on the tail of exsheathed (2% hypochlorite-treated) L3 was also evaluated [31].

### 2.6. In Vitro Tests: Egg-Hatch Test (EHT) and Larval-Exsheathment Inhibition Assay (LEIA)

The EHT was performed according to the method described by Coles et al. [32]. Using 24-well cell culture plates (TC Plate 24 Well, Standard, F, SARSTEDT S.r.l., Verona, Italy), about 100 purified eggs were placed in each well with 1 mL of a solution containing different concentrations (2400, 1200, 600, 300 and 150 µg/mL) of the tested compounds obtained from the stock solution (4800 µg/mL) prepared in phosphate-buffered saline (PBS, VWR^®^ Chemical, Solon, OH, USA). Each concentration tube was previously sonicated in a distilled water bath. For positive controls, a stock solution of 1000 µg/mL was prepared with thiabendazole (TBZ) and distilled water. For negative controls, a solution of eggs with PBS was used to keep the pH of the medium as close as possible to 7.0. Plates were incubated at 26 °C in darkness and 80% humidity while sealed with paper film, and checked after 48 h under an inverted microscope. The assay was conducted in triplicate.

The number of unhatched and hatched eggs was calculated for each well by using the following formula: number of eggs/(number of L1 + number of eggs) × 100

The LEIA was performed according to the method described by Jackson and Hoste [33]. The stock solutions were prepared as assessed for the EHT, and dilutions were performed in order to obtain 2400, 1200, 600, 300 and 150 µg/mL concentrations. The tubes containing the different concentrations were sonicated in a distilled water bath. Then, 1 mL of L3 larvae (about 1000 larvae) were incubated for 3 h at 22 °C and regularly shook with 1 mL of each of the two different sea buckthorn berry juices at each dilution (treated larvae) or with 1 mL of PBS (untreated larvae). After incubation, control and treated tubes were centrifuged at 1500 rpm for 3 min for at least 2 times, and 1 mL of supernatant was removed. Then, an optimum concentration of Milton^®^ solution (Inibsa Laboratorios, Barcelona, Spain) composed of 1% sodium hypochlorite and 16.5% sodium chloride, was added to each tube, and ensheathed and exsheathed larvae were counted at 0, 20, 40 and 60 min (T0, T20, T40, T60) after adding a drop of Lugol solution. The assay was conducted in quadruplicate. 

The percentage of ensheathed larvae (%Ensh) was determined using the following formula:%Ensh = (L3 ensh)/L3t × 100
while the percentage of exsheathed larvae (%Exsh) was calculated using this second formula:%Exsh = 100-L3ensh/L3t × 100
where L3t = total number of larvae (ensh + exsh); L3ensh = number of ensheathed larvae; and L3exsh = number of exsheathed larvae.

### 2.7. Addition of Polyvinylpolypyrrolidone (PVPP) to the Extracts

With the aim to better understand the influence of polyphenols on the *in vitro* AH effects of the tested extracts, polyvinylpolypyrrolidone (PVPP) was added to the two juices. Polyvinylpolypyrrolidone is a commercially available material produced by cross-linking polyvinylpyrrolidone [26]. This substance is water-insoluble but extremely hydrophilic, and binds water through its carbonyl functionalities via hydrogen bonding. Hydrogen bonding is the main mechanism by which PVPP binds flavonoids and a variety of other phenolic compounds. Therefore, if after PVPP exposure a loss of the anthelmintic activity is observed, it can be assumed that polyphenols are most probably responsible for the anthelmintic activity [34]. To ascertain the role of the polyphenols in the inhibiting effects on GIN egg hatching and larval exsheathment of the two examined SB berry juices, the extract solutions, at concentrations of 1200 µg/mL (for LEIA) and 2400 µg/mL (for EHT), were mixed in a ratio of 1:50 with PVPP, incubated for 2 h with regular shaking, and centrifuged at 3000 rpm for 3 min. Then, the supernatant was collected and added to L3 larvae or to eggs by using the same method as previously described for the LEIA and EHT, respectively. A negative control (PBS) and extracts without PVPP at 1200 μg/mL (for LEIA) and at 2400 μg/mL (for EHT) were also evaluated for comparison [34,35].

### 2.8. Statistical Analysis

The obtained results were used to determine the concentration required to inhibit 50% of eggs hatching and larval exsheathment (EC_50_) with respective 95% confidence intervals (95% CI) using Polo Plus 1.0 software [36]. The differences between the EC_50_ of each extract were determined by comparing the respective 95% CI. A result was considered significant when there was no overlap between the 95% confidence limits of the EC_50_ values [37,38]. For evaluating the differences between SBT and SBF, both in EHT and LEIA, a one-way ANOVA test with a post hoc Tukey honestly significant difference (HSD) test was used for comparing multiple concentrations of extract of each juice. Data were considered significant if the *p*-value was less than 0.05 (*p* < 0.05). 

Results from the assays with and without PVPP were analysed using the Student’s *t*-test, and differences were considered significant at *p* < 0.05. 

Data were analysed using SAS^®^ (Statistical Analysis Software, SAS/STAT 9.3).

## 3. Results

### 3.1. Chemical Analysis

#### 3.1.1. Total Polyphenols Content (TPC)

The TPC was 6260 ± 20 mg of gallic acid equivalents (GAE)/L (mean value of three measurements) for the sea buckthorn juice from Tuscany (SBT) and 3940 ± 20 mg GAE/L (mean value of three measurements) for the sea buckthorn juice from Friuli-Venezia Giulia (SBF).

#### 3.1.2. HPLC and HPLC-MS Analyses

Structures of the standard compounds are reported in Figure 1. All standard compounds were separated in the conditions used (using wavelengths at both 280 nm and 360 nm); the retention times and the exact masses are reported in Table 1. 

Figure 2 shows the HPLC chromatograms of sea buckthorn juice from Tuscany and from Friuli-Venezia Giulia at 360 nm.

As can be observed in Figure 2, the two profiles are very similar, indicating that the flavonoid composition was very close in both SB juices, although the intensities of the individual peaks were different. To assign the correct identification of phenols present in the juice, since not all compounds are commercially available, HPLC-MS analyses were also performed.

Table 2 reports the values for accurate mass found in the HPLC-MS analysis. Peak 1 had a mass spectrum with a base peak of 787.2327 *m/z* and a secondary peak of 463.1246 *m/z*, which could correspond to isorhamnetin-3-sophoroside-7-rha **9** (787.2297, M + H). Peak 2 had a mass spectrum with a single peak of 625.1782 *m/z*, which corresponded to either isorhamnetin-3-*O*-rutinoside **11**, isorhamentin-3-*O*-neohesperoside **12**, or isorhamentin-3-*O*-glu-7-rha **10**, since they all had the same M + H value (625.1769). Peak 3 had a mass spectrum with a base peak of 303.0511 m/z and a secondary peak of 465.1051, which corresponded to quercetin-3-*O*-galactoside **13** or quercetin-3-*O*-glucoside **14**, which had the same M + H value of 465.1033 m/z. Peak 4 had a mass spectrum with a base peak of 625.1776 and two secondary peaks of 479.1198 and 317.0654, which corresponded to either isorhamentin-3-*O*-rutinoside **11**, isorhamentin-3-*O*-neohesperoside **12**, or isorhamentin-3-*O*-glu-7-rha **10**, as already observed for peak 2. Peak 5 had a mass spectrum with a base peak of 317.0646 m/z and two secondary peaks of 479.1191 (M + H) and 501.1024 (M + Na), which corresponded to isorhamnetin-3-*O*-glucoside **4**. The structures of the identified compounds are shown in Figure 3.

Based on these analyses, it seemed that isorhamnetin, present in different conjugated forms with glucose or other carbohydrates, was the most abundantly found polyphenol in the sea buckthorn juices. The quantification of compounds corresponding to peaks 1, 2, 4, and 5 was possible using a calibration curve of isorhamnetin (see Section 2.3), while the quantification of compounds corresponding to peak 3 was conducted by using a calibration curve for quercetin (see Section 2.3). The results are reported in Table 2.

### 3.2. Egg-Hatch Test (EHT) and Larval-Exsheathment Inhibition Assay (LEIA)

The results of the EHT and LEIA of the five different concentrations of the two SB berries are presented as the mean of percentage of egg-hatch inhibition and larval-exsheathment inhibition ± standard deviation (Table 3). In the EHT, no statistically significant differences (*p* > 0.05) were found between SBT 2400 µg/mL and the reference drug (TBZ), while the results for all the other tested concentrations were statistically different only when compared to negative controls (PBS) (Table 3). Regarding the SBF, no concentration in the EHT showed an efficacy statistically comparable to that of TBZ (Table 3). However, egg inhibitions performed using the higher SBF concentrations (2400, 1200 and 600 µg/mL) were significantly different from that observed for the untreated controls (PBS) (*p* < 0.05). Lower concentrations showed a percentages of egg inhibition comparable to that of the untreated controls (PBS) (*p* > 0.05).

Regarding the LEIA, statistically relevant differences were observed for 2400, 1200 and 600 µg/mL concentrations of both SB berry juices when compared to the untreated controls (PBS) (*p* < 0.05).

A comparison between the inhibition effects of SBT and SBF on GIN eggs (EHT) and larvae (LEIA) is graphically represented in Figure 4 and Figure 5, respectively. 

In the EHT, SBT was significantly more effective than SBF in inhibiting the egg hatch at 2400, 1200 and 600 µg/mL (Figure 4). No significant differences in the larval-exsheathment inhibition activity were observed between the two SB juices at all tested concentrations (Figure 5).

The EC_50_ of SBT and SBF observed in the EHT and LEIA, as well as the relative 95% confidence limits, are reported in Table 4. A significant difference in the EC_50_ between the two tested juices was observed only in the EHT, since no overlap was observed between their 95% CI.

### 3.3. Effect of the Addition of Polyvinylpolypyrrolidone (PVPP) Treatment on Extracts’ Efficacy

The effects (mean ± standard deviation) of PVPP addition on the efficacy of the two SB juices are reported in Table 5. No significant differences (*p* > 0.05) were observed between the AH activity of the juices with and without PVPP, except for SBT in EHT, the activity of which was significantly lower after PVPP addition (*p* = 0.009, *p* < 0.05). 

Larval-exsheathment percentages observed at T0, T20, T40 and T60 (minutes) for the two juices at 1200 μg/mL with and without PVPP addition and for the negative controls (PBS) are shown in Figure 6.

### 3.4. GIN Third-Stage Larvae (L3) Identification

L3 identification showed that *Haemonchus contortus* (70%), *Trichostrongylus* spp. (20%), and *Oesophagostomum* spp. (10%), were present in the faecal pools used in the assays. 

## 4. Discussion

*H. rhamnoides*, also known as sea buckthorn, is considered a very promising plant, as it is a rich source of valuable bioactive components [18,19]. Several studies have investigated its broad biological properties, and its uses range from the medical and cosmetic fields to the industrial food sector [18,19,39,40]. The whole plant is rich in bioactive antioxidant and immunomodulating compounds, and *H. rhamnoides* berries, leaves, and seeds are widely used for both nutraceutical and medicinal purposes [18,19,22,41]. 

In the context of the search for natural compounds that can be used for the control of GINs in small ruminants, this plant was selected and tested for evaluating its anthelmintic properties *in vitro*, while also considering the presence in its composition of chemical classes of compounds, especially polyphenols, the antiparasitic activity of which has been previously reported in the literature [16,42,43]. Moreover, the juices tested in this study are commercialized for human consumption, ensuring the low toxicity of these products. 

Two different commercial juices of *H. rhamnoides* were tested *in vitro* for the evaluation of their potential anthelmintic properties. Contextually, their chemical profiles were characterized to identify their main constituents.

The two evaluated *H. rhamnoides* berry juices were obtained from plants grown in different areas of the Italian territory: one in the northeast of Italy, in the Friuli-Venezia Giulia region (SBF); and one in Tuscany (SBT), a region of central Italy characterized by a more temperate climate. Two *in vitro* assays were used to assess and compare the anthelmintic activities of these two sea buckthorn berry juices, the egg-hatch test (EHT) and the larval-exsheathment inhibition assay (LEIA) [32,33]. Using the first assay, the abilities of the two different extracts to inhibit the hatching of GIN eggs were evaluated, while the second test was performed to assess the abilities of the substances to inhibit larval exsheathment, which represents a crucial step in the GIN life cycle [44,45]. In fact, without the loss of the cuticle, it is not possible for L3 ingested by the host to penetrate the gastrointestinal mucosa, proceed in their development, and complete their life cycle [44,45]. Moreover, the LEIA and EHT also were chosen because they are considered to be highly reproducible and sensitive assays compared to other biological tests [46,47]. Furthermore, the EHT was chosen to also evaluate the *in vitro* anthelmintic activity of the tested compounds on a parasitic stage other than the larvae, considering that eggs have a better ability to withstand many adverse environmental conditions [48]. 

In the EHT, the obtained results were discordant between the two SB berry extracts. A higher egg-hatch inhibition was observed for the highest concentration of SBT (94%) compared to SBF (52.99%). The results obtained for SBT were comparable to those observed for the thiabendazole-treated positive controls used in this test. Regarding the activity against larvae, in the LEIA, both extracts were able to inhibit the larval exsheathment completely (100%, SBF) or almost completely (97.7%, SBT) at 1200 µg/mL According to Vercruysse et al. [49], the efficacy of an anthelmintic product is ensured when it has a percentage efficacy of at least 90%. Therefore, it can be stated that both SB juices were endowed with high anthelmintic activity against larvae (L3), and only SBT also against eggs. Moreover, the SBT tested in this study (EC_50_ = 519 µg/mL) was 24.7 and 3.8 times more potent in inhibiting GIN egg hatching, respectively, compared to ethanolic extracts from *Mentha pulegium* (EC_50_ = 12,800 µg/mL) [50] and *Artemisia campestris* (EC_50_ = 2000 µg/mL) [51], which were evaluated in previous studies. 

The main bioactive compounds reported for their anthelmintic activity against GINs are polyphenols, particularly condensed tannins and flavonoids, although other chemical classes of compounds have also been reported [52]. Polyphenols constitute a wide class of compounds, and according to most of the literature, they are divided into phenolic acids, lignans, stilbenes, and flavonoids [53]. The total polyphenols content of sea buckthorn juice from fruits of plants grown in Poland was previously determined by Novack et al. [54] to be 4784 ± 35 mg GAE/L. The values we obtained for the SBT and SBF were in line with the reported values. The identification of polyphenols was performed using quercetin, isorhamnetin, isorhamnetin-3-glucoside, myricetin, rutin, epicatechin, gallocatechin, and kaempferol as standards, considering that they are included among the most important compounds of sea buckthorn berries and leaves [55]. In the SBT and SBF we examined, several peaks were present in the same range of minutes in some cases; therefore, the assignment of the correct identification to standard compounds was challenging, and an HPLC-MS was also performed to confirm the identity of the compounds. Moreover, a single standard compound was used for the quantification of isorhamentin-similar molecules, considering that the glycosides bonded to isorhamentin do not affect the absorbance at UV at 360 nm. The use of a single standard compound to refer to similar compounds is in fact common [56]. Moreover, not all compounds are always commercially available to be used as standards.

Generally, sea buckthorn is rich in flavonoid glycosides, including I-3-*O*-rutinoside, I-3-*O*-glucoside, Q-3-*O*-rutinoside, Q-3-*O*-glucoside, I-3-*O*-glucoside-7-*O*-rhamnoside, K-3-*O*-sorphoroside-7-*O*-rhamnoside, I-3-*O*-sorphoroside-7-*O*-rhamnoside and rutin [53,54,55]. Flavonoids form a major subgroup within phenolic compounds in sea buckthorn, and their total concentration in berries may range from 1680 to 8590 mg/kg [57], which is a concentration several times higher than in other flavonoid-rich plants such as blackberries or dog rose [58]. The most common aglycons present in berries are kaempferol, quercetin, myricetin, and isorhamnetin [59,60,61,62]. In this study, isorhamentin-3-rutinoside, isorhamentin-3-neohesperoside or isorhamentin-3-glu-7-rha, quercetin-3-galactoside, and quercetin-3-glucoside were the most represented compounds. The results obtained in this study were therefore in line with those of Guo et al., Teleszko et al., and Heinäaho et al. [59,60,61]. These results also agreed with previously reported studies that found that flavonol glycosides were the most abundant class of phenolic compounds in sea buckthorn, particularly isorhamentin glycosides [56,60]. However, while the chemical analyses of SBF and SBT showed that both were characterized by a very similar biochemical profile and flavonoid composition, the intensities of the single peaks, e.g., the concentrations of each compound, were different. 

The anthelmintic activity of some of the compounds identified in the juices were already reported on *Haemonchus contortus*, a nematode considered to be the most relevant GIN species in small ruminants [52], and which was found to be prevalent among the GIN genera/species identified in this study. Among these compounds, the flavonol narcissin (isorhamnetin-3-rutinoside) was included among the main active compounds from sainfoin, causing a significant reduction in the motility of the larvae of the GIN species *H. contortus* [42]. Nematocidal activity against *H. contortus* eggs and larvae of isorhamnetin from *Prosopis laevigata* leaves was reported by Delgado-Núñez et al. [43]. Moreover, alterations of the cuticle of *H. contortus* L3 larvae, but no significant alterations of eggshell surface, were observed after exposure to isorhamnetin [43]. Therefore, the high inhibition effects on larvae of both juices, as well as the high inhibition of egg hatching observed for SBT, could be associated, at least in part, with the presence of isorhamnetin-3-rutinoside and other isorhamnetin glycosides in both extracts. On the other hand, differences in the *in vitro* anthelmintic activity observed for the two extracts probably could be ascribed to the different concentrations of isorhamnetin glycosides as well. 

Quercetin-3-galactoside and quercetin-3-glucoside were also identified in both extracts. Therefore, we supposed that these compounds can play a role in their *in vitro* anthelmintic effects. In support of this, it was observed that a crude ethanol extract of the aerial parts of *Artemisia campestris* capable of inhibiting egg hatching and causing death and paralysis in adults of *H. contortus* predominantly contained derivatives of the flavonol quercetin and the flavone apigenin [52]. The inhibition of GIN L3 motility by quercetin was reported by Giovanelli et al. [63]. Similarly, the inhibition of *Trichostrongylus* spp. L3 motility was reported for quercetin-3-*O*-glucopyranoside from *Vicia pannonica* [64]. Klongsiriwet et al. [65] found that the flavonoid quercetin was very efficient in inhibiting *H. contortus* larval exsheathment, even at a very low concentration (250 µM). 

With the aim to better understand the influence of polyphenols on the *in vitro* AH effects of tested compounds, PVPP (polyvinylpolypyrrolidone) was added to the two juices. This substance is water-insoluble but extremely hydrophilic, and binds water through its carbonyl functionalities via hydrogen bonding. Hydrogen bonding is the main mechanism with which PVPP binds flavonoids and a variety of other phenolic compounds. Therefore, if after PVPP exposure a loss of the anthelmintic activity is observed, it can be assumed that polyphenols are most probably responsible for the anthelmintic activity [35]. In this study, the addition of the PVPP did not result in a significant diminution of the anthelmintic efficacy of the juices *in vitro*, except for SBT in the EHT. Doner et al. [26] observed that within each class of flavonoids, binding increased with the number of hydroxyl groups, with the exceptions of kaempferol, and consequently, the derivatisation of hydroxyl groups resulted in greatly diminished binding. Moreover, they found that the effect of glycosylation on binding was especially strong, since not only was hydrogen bonding to PVPP via a potential hydroxyl functionality blocked, but access to PVPP by other hydroxyls in the molecule also was hindered [26]. In another study [66], it was also highlighted that a higher amount of PVPP should be used to bind rutin with respect to catechin, showing that the presence of the disaccharide rutinose attached to rutin caused a steric hindrance. Therefore, the lack of anthelmintic reduction after PVPP exposure observed in this study can be ascribed to the high content of glycosylated compounds in the two sea buckthorn juices, in accordance with data from previous reports [42,67]. Another possible explanation could be that the ratio of PVPP was insufficient to cope with the high phenolic content of the extracts [35,66]. These results may also imply that the standard concentration of PVPP generally used in studies evaluating the anthelmintic properties of plant compounds, as in this study, may not be enough to ascertain the role of the polyphenolic fraction in the anthelmintic activity of a plant extract. However, the possibility that other secondary metabolites also could be involved in the observed anthelmintic effects of sea buckthorn juices examined here cannot be completely ruled out.

Little is known about the mechanisms by which flavonoids can exert AH activity, as most of the available studies specifically investigated tannins and their mechanisms of action, rather than flavonoids [16]. Other than on larvae, flavonoids also have strong anthelmintic effects on eggs, inhibiting egg hatching, unlike most condensed tannins [16]. Flavonoids are internalised by worms efficiently, followed by strong metabolisation, and might therefore act against molecular targets inaccessible to condensed tannins [68]. On the other hand, it was suggested that flavonoids act through mechanisms similar to those of tannins [42]. More specifically, they may bind to the nematode shell, sheath, and cuticle, causing the blockage of osmotic exchanges and loss of flexibility, and interacting with proteins responsible for the egg hatch [16,69]. Moreover, glycosylation seems to be very important in promoting flavonoid bioavailability in the nematode [69]. Quercetin-3-*O*-glucoside is most likely absorbed from the intestine via active transport and is deglucosylated to the aglycone quercetin by β-glucosidases, showing a lower activity than the respective glycoside on *C. elegans* [70]. On the other hand, β-glucosidase-deficient parasites seem to be able to tolerate a higher amount of quercetin-3-*O*-glucoside. This fact may suggest that glycosylation is required for maximum systemic bioavailability but must be followed by deglycosylation to exert the full anthelmintic effect [16,71]. This further supports the high efficacy of *H. rhamnoides* juices tested here in inhibiting the egg hatch and the larval exsheathment of GINs, since the flavonoids identified in the extracts were in glycosylated form.

## 5. Conclusions

In this study, the anthelmintic activity of *H. rhamnoides* was reported for the first time, adding to the already numerous bioactive properties of this plant. The polyphenols identified in higher concentrations in the two *H. rhamnoides* juices investigated in this study were glycosylated forms of isorhamnetin and, to a lesser extent, quercetin. Therefore, the observed anthelmintic properties can be ascribed to the high content of these glycosylated compounds in the juices, although the identified flavonoids may not be the only compounds responsible for the anthelmintic activity. These results are promising regarding the use of sea buckthorn berry juice as a potential tool for the control of gastrointestinal nematodes in small ruminants.

## Figures and Tables

**Figure 1 biology-11-00825-f001:**
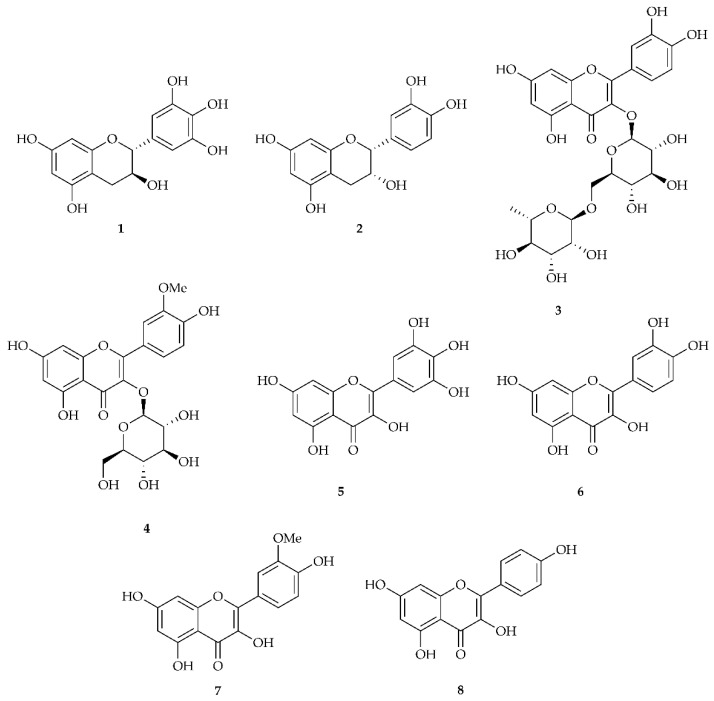
Structure of compounds gallocatechin **1,** epicatechin **2**, rutin **3**, isorhamnetin-3-*O*-glucoside **4**, myricetin **5**, quercetin **6**, isorhamnetin **7**, and kaempferol **8**.

**Figure 2 biology-11-00825-f002:**
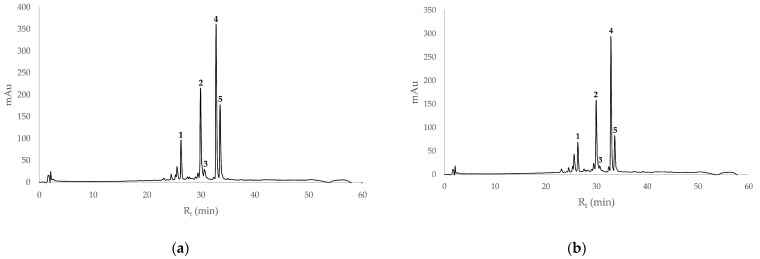
HPLC chromatogram of sea buckthorn juice from Tuscany (**a**) and from Friuli-Venezia Giulia (**b**) at 360 nm. Peak 1 had a mass spectrum with a base peak of 787.2327 *m/z* and a secondary peak of 463.1246 *m/z*. Peak 2 had a mass spectrum with a single peak of 625.1782 *m/z*. Peak 3 had a mass spectrum with a base peak of 303.0511 *m/z* and a secondary peak of 465.1051 *m/z*. Peak 4 had a mass spectrum with a base peak of 625.1776 *m/z* and two secondary peaks of 479.1198 *m/z* and 317.0654 *m/z*. Peak 5 had a mass spectrum with a base peak of 317.0646 *m/z* and two secondary peaks of 479.1191 (M + H) and 501.1024 (M + Na).

**Figure 3 biology-11-00825-f003:**
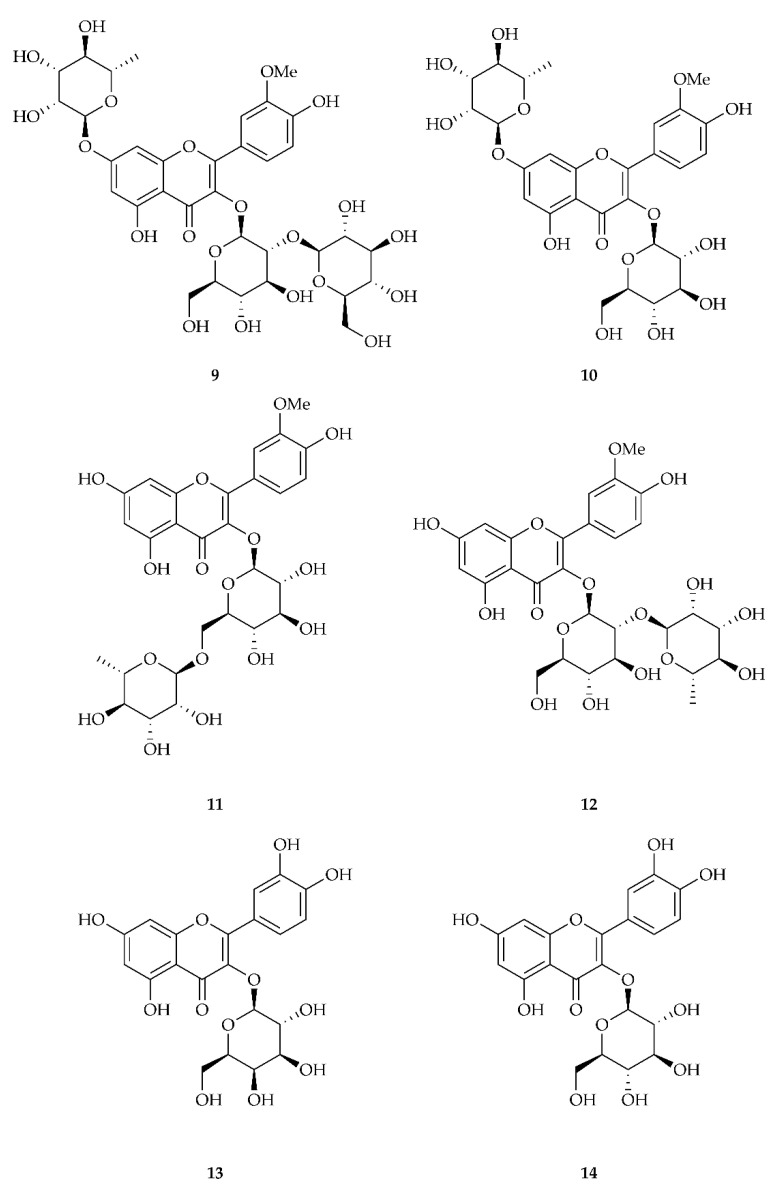
Structure of compounds isorhamnetin-3-*O*-sophoroside-7-rha **9**, isorhamnetin-3-*O*-glu-7-rha **10**, isorhamentin-3-*O*-rutinoside **11**, isorhamentin-3-*O*-neohesperoside **12**, quercetin-3-*O*-galactoside **13**, and quercetin-3-*O*-glucoside **14**.

**Figure 4 biology-11-00825-f004:**
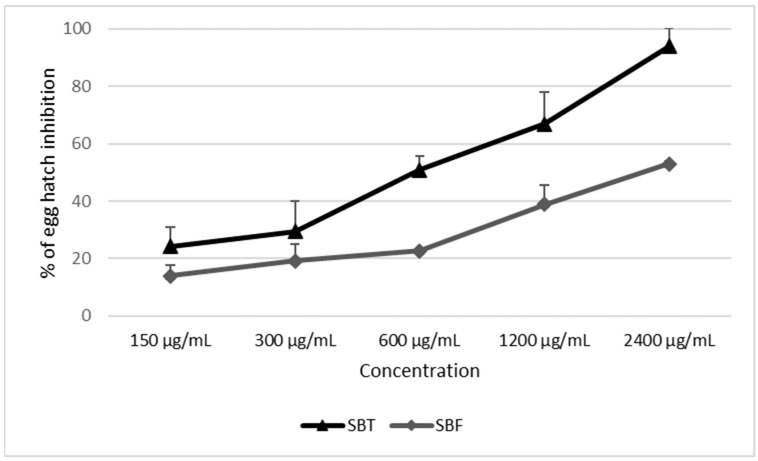
Concentration–response curves of GIN egg-hatch inhibition percentages caused by the sea buckthorn juices from Tuscany, Italy (SBT) and Friuli-Venezia Giulia, Italy (SBF).

**Figure 5 biology-11-00825-f005:**
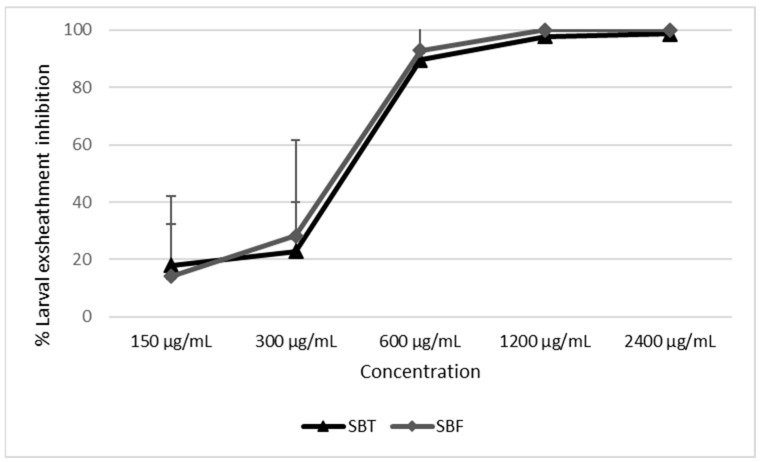
Concentration–response curves of the GIN larval-exsheathment inhibition caused by the sea buckthorn juice from Tuscany, Italy (SBT), and Friuli-Venezia Giulia, Italy (SBF). PBS: phosphate-buffered saline.

**Figure 6 biology-11-00825-f006:**
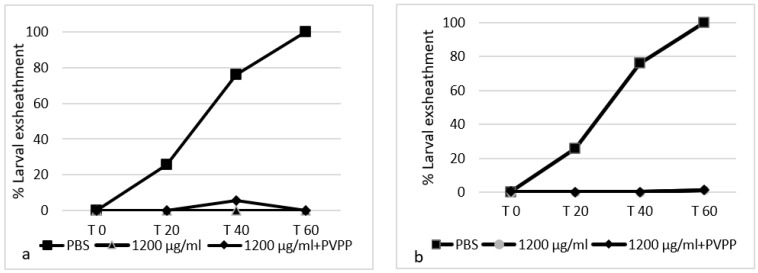
Graphical representation of the larval-exsheathment inhibition percentages caused after 0, 20, 40 and 60 min by the two sea buckthorn berry juices examined in this study (SBT and SBF) at a concentration of 1200 µg/mL, and with and without the addition of polyvinylpolypyrrolidone (PVPP). (**a**) Sea buckthorn from Tuscany, Italy (SBT); (**b**) sea buckthorn from Friuli-Venezia Giulia, Italy (SBF).

**Table 1 biology-11-00825-t001:** Retention times and exact masses of all standards compounds used.

Compound	M	M + H	M + Na	R_t_
Gallocatechin **1**	C_15_H_14_O_7_ 306.0740	307.0818	329.0637	5.98 min
Epicatechin **2**	C_15_H_14_O_6_ 290.0790	291.0869	313.0688	22.94 min
Rutin **3**	C_27_H_30_O_16_ 610.1534	611.1612	633.1432	29.97 min
isorhamnetin-3-*O*-glucoside **4**	C_22_H_22_O_12_ 478.1111	479.1190	501.1009	33.45 min
Myricetin **5**	C_15_H_10_O_8_ 318.0376	319.0454	341.0273	33.81 min
Quercetin **6**	C_15_H_10_O_7_ 302.0427	303.0505	325.0324	39.11 min
Isorhamnetin **7**	C_16_H_12_O_7_ 316.0583	317.0661	339.0481	39.95 min
Kaempferol **8**	C_15_H_10_O_6_ 286.0477	287.0556	309.0375	44.4 min

**Table 2 biology-11-00825-t002:** Accurate mass, exact mass, and concentrations (µg/mL) of each peak-correspondent polyphenolic compound found in the two the two tested sea buckthorn juices. SBT: sea buckthorn Tuscany, Italy; SBF: sea buckthorn Friuli-Venezia Giulia, Italy.

Peak	Compound	Accurate Mass *m/z*	Exact Mass	Concentration (µg/mL) SBT	Concentration (µg/mL) SBF
**1**	isorhamnetin-3-*O*-sophoroside-7-rha **9**	787.2327 463.1246	787.2297, M + H	79.8 ± 0.1	45.7 ± 0.1
**2**	isorhamnetin-3-*O*-glu-7-rha **10** or isorhamentin-3-*O*-rutinoside **11** or isorhamentin-3-*O*-neohesperoside **12**	625.1782	625.1769, M + H	269.7 ± 0.7	199.5 ± 0.4
**3**	quercetin-3-*O*-galactoside **13** or quercetin-3-*O*-glucoside **14**	465.1051 303.0511	465.1033, M + H	35.8 ± 1.5	5.0 ± 0.5
**4**	isorhamnetin-3-*O*-glu-7-rha **10** or isorhamentin-3-*O*-rutinoside **11** or isorhamentin-3-*O*-neohesperoside **12**	625.1776 479.1198 317.0654	625.1769, M + H	434.4 ± 0.7	340.8 ± 5.4
**5**	isorhamnetin-3-*O*-glucoside **4**	317.0646 479.1191 501.1024	479.1190, M + H	210.5 ± 0.7	78.8 ± 3.8

**Table 3 biology-11-00825-t003:** Percentages of egg-hatch inhibition (EHT) and larval-exsheathment inhibition (LHT) shown by the two tested sea buckthorn juices at different concentrations (±standard deviations). No statistically significant differences were found between the values indicated with the same letter. SBT: sea buckthorn Tuscany, Italy; SBF: sea buckthorn Friuli-Venezia Giulia, Italy; PBS: phosphate-buffered saline; TBZ: thiabendazole; N.A.: not performed.

Plant	Concentration (µL/mL)	Assay	
		EHT (%)	LEIA (%)
*Hippophae rhamnoides* SBT	PBS	4.44 ± 1.54 ^a^	10.61 ± 10.73 ^a^
	2400	94.00 ± 6.77 ^d^	98.60 ± 1.66 ^b^
	1200	66.89 ± 10.34 ^c^	97.72 ± 2.84 ^b^
	600	50.89 ± 4.68 ^c^	89.61 ± 3.66 ^b^
	300	29.56 ± 11.20 ^b^	22.92 ± 17.01 ^a^
	150	24.22 ± 6.34 ^b^	17.94 ± 14.32 ^a^
	TBZ	99.78 ± 0.38 ^d^	N.A.
*Hippophae rhamnoides* SBF	PBS	9.63 ± 1.24 ^a^	2.61 ± 3.21 ^a^
	2400	52.99 ± 3.94 ^d^	100 ± 0.00 ^b^
	1200	38.95 ± 5.78 ^c^	100 ± 0.00 ^b^
	600	22.6 ± 0.85 ^b^	92.86 ± 14.29 ^b^
	300	19.16 ± 6.73 ^ab^	28.29 ± 33.38 ^a^
	150	13.94 ± 0.88 ^ab^	14.06 ± 28.13 ^a^
	TBZ	100 ± 0.00 ^e^	N.A.

**Table 4 biology-11-00825-t004:** Effective concentrations (EC_50_, µg/mL) of the two *Hippophae rhamnoides* berry juices (SBT and SBF) causing 50% inhibition of GIN egg hatch (EHT) and larval exsheathment (LEIA) with the respective 95% confidence intervals (95% CI). SBT: sea buckthorn from Tuscany, Italy; SBF: sea buckthorn from Friuli-Venezia Giulia, Italy.

Substance	Assay	
	EHT (95% CI)	LEIA (95% CI)
*Hippophae rhamnoides* SBT	519 µg/mL (135–814 µg/mL)	402 µg/mL (196–557 µg/mL)
*Hippophae rhamnoides* SBF	>2400 µg/mL (2178–3728 µg/mL)	280 µg/mL (187–362 µg/mL)

**Table 5 biology-11-00825-t005:** Gastrointestinal nematode (GIN) larval-exsheathment inhibition (LEIA) and egg-hatch inhibition (EHT) percentages (±standard deviation) observed for 1200 and 2400 µg/mL concentrations in LEIA and EHT, respectively, of the two different sea buckthorn berry juices examined in this study (SBT and SBF) with and without polyvinylpolypyrrolidone (PVPP) addition. SBT: sea buckthorn from Tuscany, Italy; SBF: sea buckthorn from Friuli-Venezia Giulia, Italy.

Plant	Concentration (µL/mL)	EHI (%) ± S.D.	Concentration (µL/mL)	LEI (%) ± S.D.
*Hippophae rhamnoides* SBT	2400	94.00 ± 6.77	1200	100 ± 0.00
	2400 + PVPP	62.63 ± 8.37	1200 + PVPP	100 ± 0.00
*Hippophae rhamnoides* SBF	2400	52.64 ± 4.51	1200	99.66 ± 0.68
	2400 + PVPP	49.24 ± 20.60	1200 + PVPP	96.88 ± 6.25

## Data Availability

Not applicable.

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
