# Peer review of "In Vitro Anthelmintic Activity of Sea Buckthorn (Hippophae rhamnoides) Berry Juice against Gastrointestinal Nematodes of Small Ruminants"

_biology, 2022, doi:10.3390/biology11060825_

Round 1

Reviewer 1 Report

The authors present the in vitro anthelmintic properties of SBT and SBF, two sea buckthorn berry juices, which is a very interesting idea to find natural products with low toxicity that can be used to control the gastrointestinal nematodes for veterinary purposes.

I really appreciate the experimental and written style in the manuscript. A few suggestions for authors: 

Line 102, please explain the abbreviation for TFA

Line 131, "recovery and suspension of the eggs"

Line 167, "with 1 mL of the two different sea buckthorn berry juices"

Line 298, AH means anthelmintic activity? written as AA in abstract.

Reviewer 2 Report

In paragraph 2.2. it is necessary to the company that supplied you the reagents such as gallic acid, methanol, etc.

In the paragraph 2.6. you can insert the role of PVPP and why you used it in your experiments, rather than in discussions.

In Lane 255 you can insert if you used the DV as you done in Lane 261. 

In Table 4 I prefer  that you enter the unit of measure (ug/mL) in the table itself next to the numeric values rather than just in the caption of the table itself.

Reviewer 3 Report

In the study “In vitro anthelmintic activity of sea buckthorn (Hippophae rhamnoides) berry juice against gastrointestinal nematodes of small ruminants” the authors tried carry out a chemical identification of the compound in two different commercial sea buckthorn juice and evaluated its effect against egg hatch and larval exsheathment. A proper presentation of the results, including for the reference drug, did not allow a proper evaluation of the real potential of the extracts analyzed in the study for the anthelmintic activity. Find bellow the points that should be addressed, and added in the manuscript, by the authors.

  1. Line 22, please describe the reference drug.
  2. Line 49, it is “olivelle” a popular synonym to Hippophae rhamnoides? Make it clear in the manuscript.
  3. Line 138, at first appearance, please, full spell the abbreviation LEIA.
  4. In Fig 3, please revise peak numeration.
  5. Figure caption should be more informative in Fig 2 and 3. Images of Fig 2 and 3 should be merged in one figure, improving visualization of the comparison between them.
  6. How did the authors use isorhamnetin-3-glucoside to quantify four different compounds, corresponding to peaks 1, 2, 4 and 5, at the same time?
  7. Has a reference drug not been evaluated for LEIA?
  8. Results of EHI and LEI assay could be better presented as a concentration-response curve.
  9. EC50 should be evaluated for thiabendazole as well, to allow a proper comparison with SBT and SBF.
  10. The EC50 results above the highest concentration evaluated should be considered, please refer to this result as >2400 ug/mL.
  11. All the figures and tables captions should be revised, adding more information about abbreviation and statistical analyses.
  12. Please, revise the abbreviation for egg hatch assay, using EHI or EHT only. In addition, there are a lot of abbreviation make it difficult the reading.
  13. As there are literature reports that PVPP is not specific to polyphenols, but also binding to flavonoids (doi: 10.1590/S0103-50532008000800025), please elaborate this point in the discussion section, and revise the sentence in line 406, and line 24.
  14. The authors did not discuss about the GIN third stage larvae (L3) identification results.
  15. If polyphenols compounds are not related to the activity of rhamnoides, what is the other compounds that could be related to the anti-GIN activity?

Round 2

Reviewer 3 Report

The manuscript improved since the first submission, and the authors adressed all my concerns. Th amanuscript is now recommended for publication